# Repeated Sub-Concussive Impacts and the Negative Effects of Contact Sports on Cognition and Brain Integrity

**DOI:** 10.3390/ijerph19127098

**Published:** 2022-06-09

**Authors:** Michail Ntikas, Ferdinand Binkofski, N. Jon Shah, Magdalena Ietswaart

**Affiliations:** 1Department of Psychology, University of Stirling, Stirling FK9 4LA, Scotland, UK; 2Division for Clinical Cognitive Sciences, Department of Neurology, University Hospital RWTH Aachen, 52062 Aachen, Germany; 3Institute of Neuroscience and Medicine (INM-4), Forschungszentrum Jülich, 52425 Jülich, Germany; 4JARA-Brain-Translational Medicine, 52062 Aachen, Germany; 5Department of Neurology, University Hospital RWTH Aachen, 52074 Aachen, Germany

**Keywords:** contact sports, soccer, heading, traumatic brain injury, concussion, sub-concussion, brain health, dementia prevention, neuroimaging, fluid biomarkers

## Abstract

Sports are yielding a wealth of benefits for cardiovascular fitness, for psychological resilience, and for cognition. The amount of practice, and the type of practiced sports, are of importance to obtain these benefits and avoid any side effects. This is especially important in the context of contact sports. Contact sports are not only known to be a major source of injuries of the musculoskeletal apparatus, they are also significantly related to concussion and sub-concussion. Sub-concussive head impacts accumulate throughout the active sports career, and thus can cause measurable deficits and changes to brain health. Emerging research in the area of cumulative sub-concussions in contact sports has revealed several associated markers of brain injury. For example, recent studies discovered that repeated headers in soccer not only cause measurable signs of cognitive impairment but are also related to a prolonged cortical silent period in transcranial magnetic stimulation measurements. Other cognitive and neuroimaging biomarkers are also pointing to adverse effects of heading. A range of fluid biomarkers completes the picture of cumulating effects of sub-concussive impacts. Those accumulating effects can cause significant cognitive impairment later in life of active contact sportswomen and men. The aim of this review is to highlight the current scientific evidence on the effects of repeated sub-concussive head impacts on contact sports athletes’ brains, identify the areas in need of further investigation, highlight the potential of advanced neuroscientific methods, and comment on the steps governing bodies have made to address this issue. We conclude that there are indeed neural and biofluid markers that can help better understand the effects of repeated sub-concussive head impacts and that some aspects of contact sports should be redefined, especially in situations where sub-concussive impacts and concussions can be minimized.

## 1. Introduction

Sports, especially when performed on a regular basis, are yielding a wealth of benefits for the cardiovascular fitness, for psychological resilience and for cognition. Because of the dose–response relation between physical activity and health, persons who wish to further improve their personal fitness, reduce their risk for chronic diseases and disabilities or prevent unhealthy weight gain may benefit by exceeding the minimum recommended amounts of physical activity. However, sport participation comes with its own risks, especially the risk of injury. Those injuries can be musculoskeletal, or brain related. Repeated concussion has been shown to pose a risk to future brain health [1], but the concern is that the low level but repeated nature of routine head impact in contact sport is similarly detrimental to otherwise healthy individuals. This is particularly relevant in the context of the current Special Issue dedicated to evidence for multifaceted positive effects of exercise and sport on cognitive functions and for prevention of cognitive decline. 

In this review we therefore focused on the negative consequences of sport participation on brain health by reviewing the literature on the effects of repetitive sub-concussive head impacts in contact sport participation. The aim of our contribution to this Special Issue is threefold. First, we aim to draw attention to possible negative effects of sport-related exposure to impacts. Secondly, we want to highlight the most promising avenues for the much-needed increase in research activity on this topic. Lastly, we warn against any complacency within sport governance in light of scientific evidence.

## 2. Potentially Negative Effects of Sports on Brain Health

Concern about the danger that sport-related concussion poses to the nervous system has become more accepted, in particular in contact sports such as American football, soccer, rugby, ice hockey and boxing. The rates of sport-related concussion are high and have been found to be increasing yearly [2]. “One population in particular, high school athletes, must be monitored more closely as research indicates that they are neurologically more vulnerable than the collegiate population” [3]. This monitoring is important especially since teenagers have been found to have the highest rates of sport-related concussion amongst athletes [2], with two thirds of concussions involving athletes under the age of 19 [4]. There were approximately 800,000 reported concussions suffered by high school athletes in the US between 2005 and 2010, with boys’ American football accounting for 47.8% of this number [5]. However, the examination of the incidence of traumatic brain injury at scale dated back at least 20 years, until the launch of a recent large-scale international observational study [6]. Cohort studies such as this are critical to examine sport-related brain injury incidence, sport-specific profiles, demographic data such as age and sex, and critically the incidence and outcomes of those incurring multiple brain injuries while participating in sport. While this monitoring of concussions is important for the prevention of further catastrophic damage, in particular that associated with second impact syndrome [7], a number of studies have also demonstrated that neurophysiological changes can accumulate due to repetitive sub-concussive blows [8,9,10,11,12,13,14]. As yet, the physiological mechanisms underlying the consequences of sport-related impact in general, and repetitive sub-concussive head impacts (RSHI) in particular, are yet unknown. Below, we consider what the different assessment measures now available reveal about the nature of the effects of RSHI on brain health.

## 3. What Cognitive Markers Tell Us about the Consequences of Routine Impact in Sport

Literature on the effects of RSHI on cognitive markers (neurocognitive testing) is mixed, with studies showing some cognitive domains to be affected by RSHI while others failed to find any link between RSHI and neurocognitive performance [15]. The majority of neurocognitive studies resorting to clinical instruments used to diagnose and monitor full-blown concussion, such as concussion recognition tools SCAT 3–5 and ImPACT, did not find any evidence [14,16,17,18] concerning the consequences of routine (sub-concussive) head impacts in sport. However, some researchers identified differences between the reaction time and processing speed of non-concussed players in contact sports with high concussion-incidence compared to players with “low-contact sports” by using ImPACT [19].

Studies that used cognitive tests assessing specific cognitive domains such memory, attention, learning and reaction time, also yield mixed results, with some finding RSHI affecting cognition [20,21,22,23,24]. Specifically, performing 20 soccer headers, akin to a routine heading drill, was found to impair scores in a pair associative learning task compared to baseline assessment [20]. Additionally, the number of recently performed headers was found to be negatively correlated with psychomotor speed while the number of headers performed the past year was negatively correlated with verbal learning and verbal memory performance [22]. Moreover, soccer, American football, and ice hockey players’ scores on sensorimotor function and learning have been found to improve less when compared to the score improvement shown in non-contact sports athletes throughout a season of competitive play [21,24], indicating the hindering of the learning effects associated with cognitive tasks.

One major issue about the effects of RSHI on cognition is that impairment in any cognitive function is a symptom of alterations in a physiological/neurological mechanism that is yet unknown. Establishing a better understanding of what physiological mechanisms are affected by RSHI would provide a better understanding about which aspects of cognition we expect to be affected. Axonal damage, structural alterations, increased cortical inhibition, neurovascular damage, excitotoxicity, oedema, blood flow alterations, are all mechanisms that might play a role, and each of them would result in different symptomatology in the cognitive domain. Without this information at hand, cognitive studies are essentially guessing, trying to find how cognition is affected by using a wide range of tests that assess different cognitive functions without providing further insights into why those functions are expected to be impaired. Reliable gold-standard cognitive assessment, designed to identify gross cognitive deficits following significant brain damage, may furthermore lack the fine-grained resolution and sensitivity required when documenting sub-clinical effects and accumulation of subtle consequences of RSHI in relation to each of these potential mechanisms. If any gross deficits were arising from routine impact in sport, we would probably have known about it. The use of broad-range clinical instruments is therefore questionable. Furthermore, the use of multiple tests in the same studies increases the chances of false positives that can make the literature of the field incoherent and unreliable. Historically, established cognitive testing has been the standard method of evaluation of the consequences of exposure to RSHI. This may be why only neurocognitive studies are of sufficient numbers to conduct meta-analyses, for example, of adverse effects of football heading [25]. However, the lack of sensitivity of these measures could lead to misleading claims that there are no consequences to such exposure in sport.

## 4. Understanding the Negative Consequences of Sport and the Current State of Neurophysiological Biomarkers

### 4.1. Neuroimaging

One way to gain a more in-depth understanding of the effects of RSHI on athletes’ brain health is by using neuroimaging modalities. Neuroimaging biomarkers have proven to be valuable in detecting brain changes that occur before neurocognitive symptoms in contact sport athletes. Unlike cognitive markers, neurophysiological markers provide evidence of the deleterious effects that RSHI have on the athletes’ brains [15]. This relationship between sub-concussive impacts and evidence of altered brain function has been examined using multiple method modalities: structural (magnetization transfer imaging; voxel-based MRI morphometry; diffusion tensor imaging—DTI); metabolic (magnetic resonance spectroscopy—MRS); functional (fMRI; resting-state fMRI); and electrophysiological (transcranial magnetic stimulation; EEG). Below, we explore the contribution on the issue from each of these domains. As part of this, we highlight the most promising avenues in this domain. For example, we highlight the relevance of examining acute changes in the metabolic state of athletes’ brains following routine impact using magnetic resonance spectroscopy. We also highlight the potential of a more advanced MRI technique—qMRI (quantitative magnetic resonance imaging)—that offer the potential to quantitatively measure physical parameters such as the relaxation times T1 and T2* and, importantly, water content in brain tissue. Water content reflects the accumulation of oedema at the site of injury in the brain and it is noted that low-grade oedema can be easily overlooked or confused with imaging artefacts arising from imperfect hardware, for example. We also highlight the need for more EEG research, which can offer further insights on the cognitive functions and the brain alterations caused by repetitive head impacts, before we turn to another important emerging field in the context: biofluid marker research.

#### 4.1.1. Structural Imaging

A neuropathologic examination of a retired former National Football League player, who experienced several blows to his head but did not have any history of brain trauma or concussion, revealed clear signs of chronic traumatic encephalopathy (CTE) [26,27,28]. Additionally, McKee and colleagues reported 46 neuropathologically diagnosed cases of CTE (90%) among 51 athletes. The most prominent neuronal loss was seen in the hippocampus, entorhinal cortex and amygdala, with less severe losses in the subcallosal and insular cortex, olfactory bulbs, mammillary bodies, locus coeruleus, substantia nigra, medial thalamus and cerebral cortex [12]. These cases kindled the discussion about possible negative effects of contact sports on athletes’ brain health, in particular where there is high incidence of concussion. However, the effects of concussion and RSHI are conflated in all contact sports, perhaps with the exception of soccer where sport-related exposure is mainly due to routine head impact. At the time, concerns around RSHI and brain integrity was perhaps excepted somewhat following an earlier conventional structural magnetic resonance imaging (MRI) study that found no correlation between career soccer heading exposure and structural abnormalities [29]. This negative result was confirmed in one well-designed prospective study where no conventional MRI changes were discernible in professional soccer players (observation period = 5 years) and also no cognitive changes were found [30].

On the other hand, in a retrospective study of ex-professional soccer players, voxel-based MRI morphometry demonstrated cortical thinning in the right inferolateral parietal, temporal and occipital cortex and was found to be associated with worse performance on one of six cognitive tests (Rey–Osterrieth complex-figure long-delay recall) while also being inversely correlated with lifetime estimates of number of soccer headers [31]. Additionally, in college soccer players, voxel-based MRI morphometry showed decreased grey matter density and volume within the anterior–temporal cortex [32].

In an earlier study of professional soccer players, DTI indicated widespread white matter abnormalities (albeit no changes in fractional anisotropy) [33]. However, Lipton and colleagues found lower levels of fractional anisotropy in parieto–occipital areas which was associated with decreased scores in one of six cognitive tests (poorer memory) and was inversely correlated with the annual number of headers [34]. All those findings are bound to a longer history of header-associated routine head impact and signal additive effects of sub-concussion on the white matter. In a more recent study, Manning et al. [35] found in a female group of contact athletes with and without concussion over a season an increase in mean axial and radial diffusivity with decreased fractional anisotropy in multiple white matter tracts. Axial diffusivity was significantly lower in the genu and splenium of the corpus callosum in those contact athletes with a history of concussion. 

Of note is the potential protective effect of wearing a collar that compressed the jugular vein [36]. In a group of female athletes, the researchers found that significant white matter changes in DTI parameters were detectable only in the non-collar group, whereas there were no changes in the collar group. Measures such as this may lessen the burden of concussion and RSHI in active contact sports athletes, but more research in this area is necessary to confirm the effectiveness of such protective measures. However, if proving reliable, the efficacy of a blood-flow-related intervention does also have the potential to reveal which mechanisms are underlying the effects of unprotected sport-related head impact, which would advance neurological understanding.

Perhaps unsurprising, DTI changes do not seem to present reliably; for example, in a pilot study on 10 athletes Kenny et al. [37] did not find any changes in fractional anisotropy or mean diffusivity after one round of heading the ball. An informative systematic review of DTI in contact sport athletes can be found in Schneider et al. [38].

In conclusion, despite the lack of sensitivity that MR-based structural neuroimaging may present, an association has been found between routine sport-related head impact, and structural abnormalities that appears dissociable from concussion and that is indeed indicative of cumulative negative effects of RSHI in contact sport leading to structural alterations of white matter.

#### 4.1.2. Magnetic Resonance Spectroscopy of the Brain

Important information about the metabolic state of athletes’ brains comes from proton (H1) magnetic resonance spectroscopy (MRS) investigations. Poole and colleagues performed a comprehensive and prospective H1 MRS study on a cohort of high school American football athletes [14]. They studied the following metabolites: N-acetyl aspartate (NAA)—biomarker of neuronal integrity; glutamate and glutamine (Glx)—a neurotransmitter and its precursor, both related to synaptic activity; creatine-containing compounds (tCr)—a mirror of energy metabolism; choline-containing compounds (tCho)—markers of membrane turnover; and myo-inositol (Ins)—an osmolyte involved in glial cell growth. In contact sport athletes, changes in neurometabolism within the dorsolateral prefrontal cortex and primary motor cortex along with changes in neurocognitive performance were found. A reduction in total creatine (tCr) and in dorsolateral prefrontal inositol (Ins) on the one hand, and deviations in the glutamate and glutamine (Glx) concentration in primary motor cortex on the other, were observed [14]. Long-term effects of high exposure to RSHI on neurochemistry in athletes without a history of clinically diagnosed concussion were investigated by Koerte et al. [39]. Compared with control athletes, soccer players showed a significant increase in both choline (Cho) and myo-inositol (mL). Additionally, ml and glutathione (GSH) were significantly correlated with lifetime estimate of RSHI within the soccer group. In contrast to Poole et al. [14], there was no significant difference in neurocognitive tests between groups. Both studies demonstrate that RSHI can have significant negative effects on brain metabolism, which precede cognitive deficits [14,39].

Interestingly, a study by Panchal et al. [40] aimed at examining exposure to concussive and sub-concussive repeated head impact, compared H1-MRS and neurocognitive evaluation before and after the Canadian Interuniversity Sports ice hockey season in male and female athletes. Over the season they found a significant decrease in NAA concentration and a trend towards a decrease in Cho concentration in the corpus callosum. Although in females there was a trend towards a decrease in glutamine (Glu) and in males an increase in Glu, in both males and females, a negative correlation was observed between changes in Glu and changes in verbal memory. Another more quantitative study was performed by Bari et al. [41]. In this study, paired H1-MRS with head impact monitoring was performed to quantify the relationship between metabolic changes and head acceleration event characteristics in high school-aged male American football players and in female soccer players. During the period of exposure to sub-concussive events, asymptomatic male American football players exhibited statistically significant changes in concentrations of glutamate+glutamine (Glx) and total choline-containing compounds (tCho) in dorsolateral prefrontal cortex, and female soccer players exhibited changes in glutamate+glutamine (Glx) in primary motor cortex. Neurometabolic alterations observed in the male American football players during the second half of the season were found to be significantly associated with the average acceleration per head acceleration events, being best predicted by the accumulation of events exceeding 50 g-force [41].

Although to date only few studies exist that have employed spectroscopy to examine metabolic state of athletes’ brains, all these findings indicate possible glial dysfunction, alteration of neural signalling and of energy supply to the brain due to cumulative effects of RSHI.

#### 4.1.3. Functional Magnetic Resonance Imaging (fMRI)

Functional neuroimaging also yields important insights into the effects of collision events on brain function. In a prospective study, neurological performance and health following head collision events was assessed in small cohort of 11 high school American football players, using longitudinal measures of collision events (HIT-System), neurocognitive testing (ImPACT) and fMRI (N-back memory task) [13]. Athletes with history of concussion, and with cognitive symptoms, had altered activation in the left middle and superior temporal regions in comparison with those without concussion and cognitive symptoms. Interestingly, the group with no history of concussion but with cognitive symptoms had altered activation in the left dorsolateral prefrontal cortex. The latter group had significantly higher numbers of non-concussive head contact events to the top-front of the head, directly above the dorsolateral prefrontal cortex, a structure important for working memory [13].

A study by Svadi and colleagues prospectively paired functional magnetic resonance imaging with head impact monitoring, to track cerebrovascular reactivity changes throughout a season and to test whether the observed changes could be attributed to mechanical loading experienced by 14 female athletes participating in high school soccer [42]. Breath-hold fMRI revealed significant reductions in frontotemporal cerebrovascular reactivity persisting up to 4–5 months after the season had ended. A similar reduction in cerebrovascular reactivity had previously been observed in a large group of patients with actual mild traumatic brain injury who were investigated with functional transcranial Doppler sonography of the middle cerebral artery [43]. Similarly, Len and colleagues found a reduction in cerebrovascular reactivity following sport-induced concussion using transcranial Doppler of the middle cerebral artery and PETCO2 measurements [44] (for review, see [45]).

Using resting-state fMRI, Manning et al. [35] followed a group of 73 female non-concussion contact sport (rugby) athletes over a season and found modulated functional connectivity in default mode and visual networks, which was related to changes in white-matter structures.

A large-scale prospective study in male and female collegiate American football players, characterized the time course of the acute physiological effects of sport-related concussion as measured with resting-state fMRI and their relationship with clinical recovery [46]. Here also, concussed athletes had elevated local connectivity in the right middle and superior frontal gyri at the 24 h visit that returned to normal levels by the asymptomatic and post return-to-play visits. Moreover, elevated local connectivity in the middle and superior frontal gyri at the 24 h visit in concussed athletes was associated with elevated psychological symptoms at the time-point at which athletes were cleared to start their return to play progression. These results suggest that sport-related concussion is associated with an acute alteration in local functional connectivity that follows a similar time course as clinical recovery in athletes who ultimately recover [46].

In general, functional imaging demonstrates changes in activity and in functional connectivity, which are tightly related to cognitive symptoms and can partially explain them. Moreover, significant changes to the cerebrovascular reactivity signal a more global reaction of the central nervous system to RSHI.

#### 4.1.4. Quantitative Magnetic Resonance Imaging (qMRI)

As already noted above, neuroimaging has established itself as a powerful tool for investigating brain-related sports injuries. Nonetheless, in contrast to *qualitative* imaging, as used in most investigations, *quantitative* MRI yields precise and accurate data, which are, for a given field strength, hardware-independent and enable objective interpretation as well as numerical comparison. These methods allow for sensitive detection of even slight concussion-related pathologies at early stages after impact.

The development of accurate methods for quantitative mapping of T1, as well as free water content has a long and successful history [47,48,49]. Advanced MRI techniques that enable the quantitative measurement of physical quantities such as water content (oedema) and relaxation times are also being used with increasing frequency [47,49,50,51,52,53,54,55,56,57,58,59,60,61]; [48,51,62,63,64]. Quantitative measures of T1 have long been used as a surrogate for water content in different pathologies, which can be measured quantitatively [47,49,50,51,52,53,54,55,56,57,58,59,60,61]. Moreover, very recently, the first quantitative water content atlas of the healthy human brain has been published [51] facilitating future quantitative comparisons of the brains of individuals with suspected brain injury with a normal cohort. Oedema, in the form of elevated water content will thus be visible in such comparisons and, importantly, the progression or regression of oedema can thus be studied with the aim being that, in the future, water content can be used to extract prognostic value of concussion related changes from qMRI.

A recently published method facilitates three-dimensional quantitative imaging of the free water content, T1, and the transverse relaxation time (T2*) [49]. This method can be easily extended to enable the measurement of the semi-quantitative magnetisation transfer ratio (MTR), a sensitive marker for demyelination. A combination of quantitative MR parameters and MTR increases the specificity of the imaging and might help to provide more insight into the underlying mechanisms and pathology associated with sports-related head impacts.

### 4.2. EEG Studies of Repetitive Sub-Concussive Head Impacts

Despite being a long-established evaluation method, electroencephalogram (EEG) studies appear to be limited in this field, while some initial efforts decades ago found that active and former soccer players had higher rates of “abnormal” EEG recordings compared to age-matched controls [65,66]. The aforementioned studies offer only limited insight into the effects of RSHI, as there was no control of the physical exercise between the groups and the EEG recordings were visually inspected, making the findings potentially biased.

Recently, more studies using EEG to investigate if RSHI affect athletes’ brains have been reported, mainly recording event related potentials (ERPs) in tasks designed to assess attention. Active soccer players were found to have decreased ERP component P3b and P3a amplitudes compared to a control group, indicating that these soccer players had diminished attentional resource allocation and attentional orienting compared to athletes practicing sports with no head impacts [67]. This was a cross-sectional study however, therefore no direct link can be made to the effects of heading on the brain. Controlled experimental studies, examining the cause-and-effect relationship between heading and brain electrophysiology, have not yet been reported. In another study, third and fourth year collegiate American football players were also found to have diminished P3b amplitudes compared to their first-year co-athletes, indicating a measurable cumulative effect of RSHI over multiple seasons of play on brain activity and attentional resource allocation [68]. In this study, however, those changes were not visible after a single season of American football [68].

Although studies are limited in number at present, the appeal of EEG as an evaluation method is that it offers brain-based measures of cognition. To date, EEG evaluations of RSHI show that there is a potential link between head impacts and attention. However, as EEG is particularly well established in the domain of attention, and attention-related ERP components are particularly robust, the first logical step when using EEG to investigate the effect of RSHI on athletes’ brains would be examine any attention decrements caused by RSHI. However, as mentioned in Section 3 above, we do not actually know at present which aspects of cognition to target when examining the effects of RSHI. Therefore, despite EEG’s more limited sensitivity in domains outside attention, EEG methods should also be employed to investigate the consequences of RSHI on other cognitive domains such as learning and memory in future research.

### 4.3. Transcranial Magnetic Stimulation Studies of Repetitive Sub-Concussive Head Impacts

Transcranial magnetic stimulation (TMS) is another physiological modality used to assess the effects of concussion and RSHI on athletes’ brains. TMS-derived intracortical inhibition has been found to be altered after a sport-related concussion in several studies [69,70,71,72,73,74,75]. Cortical silent period (an indicator of corticomotor inhibition) has been found to be increased both at the acute stage after a concussion [70,73,75] and several months later [72]. RSHI have yet to be investigated in depth with the use of TMS, with some studies showing increased cortical silence after soccer heading, rugby tackling and boxing bouts [20,76,77]. Specifically, the cortical silent period was found to be increased after 20 soccer headers compared to baseline [20], after a 3 min sparring bout compared to a baseline assessment and a non-contact control group [76] and after 15 rugby tackles compared to baseline and compared to a control group [77].

More research is necessary on TMS and RSHI; however, when taken together with the findings from the imaging literature, TMS studies provide additional evidence that RSHI can induce alterations in brain neurochemistry similar to those seen following brain injury and potentially indicative of the mechanism underlying long-term effects of routine repetitive head impacts in sport.

## 5. Sport-Related Impact and the Current State of Fluid Biomarkers

A wide range of biomarkers that assess brain chemistry, axonal damage and neuroinflammation have been used to assess sport-related head impact so far, such as s100 calcium binding protein b (s100b), tau (either total or phosphorylated), neuron specific enolase (NSE), creatine kinase isoenzyme BB (CK-BB), neurofilament light (NfL) and glial fibrillary acidic protein (GFAP). In the past few years, the field seems to be focusing more on NfL, S100b and tau, which has resulted in mixed findings, with boxing, American football and soccer being the most studied sports [78,79,80,81,82,83].

Specifically, NfL concentrations in blood have been found to be increased in American football players after a practice session [84] and after a competitive season [85,86] compared to baseline measures. In boxers, NfL has been found to increase in corticospinal fluid and in blood after a series of boxing bouts [82,87,88]. In soccer players few studies have been reported, with the only reported randomised controlled laboratory study showing that a series of 10 headers significantly increases blood NfL concentrations compared to a control condition [78].

Neuron-specific enolase also appears to be a promising marker, with a handful of studies reporting significant increases after RSHI [89,90,91]. Tau (either total or phosphorylated), on the other hand, appears to be showing mixed results [92,93,94,95] with studies finding no tau increase immediately after 40 soccer headers [93]. Similarly, no alteration of tau concentrations was found during a season of American football [92], but increased tau concentrations were found immediately after American football practices compared to baseline assessments; however, this increase was not correlated to the amount and severity of head impacts during the practices [95].

Recent soccer studies are implementing laboratory designs [78,83]. By doing so, the effects of repetitive head impacts on fluid biomarkers can be examined without the effects of exercise or of body impacts, which can influence certain biomarker concentrations. Unlike soccer, many studies of American football and boxing, even though they provide promising findings, fail to provide a clear distinction between the effects of RSHI and body impacts or exercise.

Generally, the use of fluid biomarkers to examine RSHI is still in its infancy but is showing increased traction in the last decade with a marked increase in the number of studies aiming to examine the relationship. However, it should be noted here that some of the studies in this field present several limitations, especially since some of the biomarkers assessed can also be increased by other aspects of sport participation; for example, S100b is also increased after increased physical activity. A recent review concluded that the markers of neuroinflammation seem to be promising and future research should focus on them [15]. The relevant literature has not yet been reviewed in depth; however, efforts to map the literature are being made [96]. Fluid biomarkers in parallel with neuroimaging biomarkers can provide a clearer picture of the consequences of exposure to RSHI, explain which brain processes those impacts are inducing, and what can be the potential effects of those processes on athletes’ everyday cognitive functioning and inform the understanding of the potential long-term effects such as neurodegeneration.

Thus far, the findings from neuroimaging and biofluid marker studies are providing some evidence that RSHI affect the brains of the athletes in ways that can impair their cognitive functioning in the short-term and potentially induce changes that in the long-term can lead to neurodegeneration.

## 6. What Is the Way Forward? Research Avenues and Preventive Measures

Although it is well established that physical activity and sports participation have specific physical and mental health benefits, there is strong emerging evidence that participation in some specific types of athletic activities, namely contact sports, comes with adverse effects and that there are negative consequences to brain function and brain health.

Of course, the issue of sport-related head impacts intuitively raises concerns at a grassroots level, in particular where it concerns young people. It is also understandable that there is a resistance at a sport governing level to introduce any changes to the sport without an abundance of research, considering what is at stake. However, recent development in the understanding of the chronic and evolving neurological consequences of traumatic brain damage [1] does warrant a potential complete rethink regarding the acceptability of sport-related impact, both at a concussive and a sub-concussive (routine) level. Recent technological advancement means that it is now possible to employ sensitive and informative brain measures, and it is a matter of urgency that support is provided to apply the latest neuroscientific methods to reveal what goes on in the brain as a result of exposure to sport-related head impact. We welcome the trend towards larger, well-designed, controlled studies using the newest techniques. The outcome of this is that we obtain increasingly more reliable and better interpretable results.

What is also required is a more interdisciplinary and multimethods approach, in particular when focused on cause and effect or dose–response. As the results from single measures are easily dismissible; converging evidence instead offers both a compelling evidence-base and offers insight concerning the mechanisms at play. From the review above we can see that measurable cognitive deficits in sport can be explained through brain damage as represented by findings in structural, functional and metabolic imaging, in electrophysiological measurements and in fluid biomarkers. It is therefore important not to dismiss less traditional research approaches as “experimental” or “unproven”, but instead embrace the opportunities offered by cutting-edge neuroscience and technological innovation. We have noted above the important explanatory power of newly applied methods such as TMS, EEG, qMRI and MR spectroscopy, and how these offer a window into the nature of the potentially negative consequences of sport, in particular when employed in combination and relation to more indirect measures such as cognitive function, motor control or blood biomarkers. Today’s situation concerning fluid biomarkers is that information about the effects of sport-related head impact is still not consistent. One of the reasons is that the older studies were in many cases underpowered and had an exploratory character. However, the bottom line is that evidence for the negative effects of sub-concussions in contact sports is growing.

Combined evidence suggests that RSHI seems to alter the brain. In this review, we covered the evidence pointing to acute alterations following a series of RSHI and to cumulative effects of RSHI throughout seasons of contact sport participation. The link between those acute and sub-acute (semi-acute) effects with long-term consequences is yet to be made, but research involving retired contact sport athletes presents the case for increased risk of neurodegenerative diseases in populations without history of concussion, indicating the deleterious long-term effects of RSHI [12,97].

Raised awareness of the adverse effects of RSHI on athletes’ brains has already led to changes in policy. In February 2020, soccer’s Football Association (FA) in England, the Scottish FA and the Irish FA changed their governance policy by: (i) issuing updated heading guidance for training; (ii) stating that under-11s should no longer practice heading; (iii) severely limiting heading practice frequency for 12–16 year olds; (iv) reducing practice frequency for 16–18 year olds; along with (v) reduced ball pressures; and (vi) specific ball sizes for this practice [98,99,100,101]. In June 2020, UEFA (Europe’s governing body of soccer the Union of European Football Associations) issued guidelines for heading, to protect the health and safety of youth players, stating that the guidelines are to manage heading in training and in matches aimed at “*limiting the header burden in youth football*”, and the UEFA Medical Committee chairman noting the decision was taken “*after debates continued over whether* heading *a football could lead to altering a player’s brain*” [102]. We want to point out, however, that there is an absence of research to inform at which age it is safe to start incurring RSHI, or that reduced ball size or ball pressure mitigates the burden of RSHI. In July 2021, the FA took further action by introducing heading guidance across every level of the professional and amateur game, recommending that a maximum of 10 higher force headers are carried out in any training week. We note that although research from our laboratory confirmed brain changes after 20 headers, there is an absence of research to show that taking 10 headers is safe. Future soccer studies should aim to assess the effects of 10, e.g., versus 20 headers realistically replicating routine soccer practice, and quantify the magnitude of those headers. The need for these types of studies to investigate the cognitive, biological, structural and neurochemical changes caused by heading was also highlighted in a recent review [103]. This is a demonstration of how further research will be required to guide governing bodies seeking to impose restrictions in RSHI in sports, even before we have gained a better understanding of the source and mechanisms underlying any negative consequences of sport-related impact.

In conclusion, we reiterate that it is not currently known how much sport-related impact is safe. It is the task for future studies to work out the dose-dependent effects of sport contacts and determine whether there are, in fact, safe limits in sport-related impact.

## 7. Conclusions

The emerging evidence reviewed here suggests that routine head impacts in many popular contact sports have adverse effects, in the context of the otherwise positive effects of exercise and sport on cognitive functions and the prevention of cognitive decline discussed elsewhere in this Special Issue. Considering the potential public health “timebomb” that constitutes these adverse effects of sports such as soccer played by hundreds of millions around the globe (often from a young age), there should be a great sense of urgency in funding the best scientific research that has the power to reveal and explain the link between sport and poor brain health outcomes. To date, there appears to be little commitment to generous funding either from sport or from government. The evidence reviewed here shows that the neuroscientific community has moved to turn their innovation and fledgling research programs on the topic into important first steps in revealing the dangers of routine head impact in sport. These multimethod capabilities, when brought together in an orchestrated way and reaching across the disciplines and established traditions, can provide real answers to the public, practitioners, sports governance and government, and offer the solid evidence-base that is urgently needed to take meaningful action in addressing this public health concern.

## Data Availability

Not applicable.

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
