# Peer review of "Repeated Sub-Concussive Impacts and the Negative Effects of Contact Sports on Cognition and Brain Integrity"

_ijerph, 2022, doi:10.3390/ijerph19127098_

Round 1

Reviewer 1 Report

This is a good review of negative effects of sub-concussion in contact sports with demonstration of different measurable methods and its current scientific evidence.

I have only a minor issue:

A flow chart with the number of analyzed studies (review platform/study inclusion/exclusion/total study for the review) might be helpful to follow the manuscript better.

Author Response

We are glad the reviewer feels this is a good review. Further to this reviewer’s point we need to clarify, also noted by Reviewer 2, that this is a narrative review rather than a systematic review. As we have not analysed existing studies based on inclusion criteria as we would when we conduct a systematic review, we are sorry it is not possible to include a flow chart. We hope this is acceptable.

Reviewer 2 Report

General comment

I would like to thank the Editor for the opportunity to review this manuscript for International Journal of Environmental Research and Public Health. This is a narrative review that aimed to present the current scientific evidence on the effects of sub-concussive impacts in contact sports. I really appreciate the effort of the authors working in this field, because sport-related concussion and sub-concussions has a severe impact in teenager athletes’ health. This review is really well written and organized from a methodological point-of-view. Please, find my specific comments here below in order to improve the quality of the manuscript

Specific comments

Page 1 – Line 20. Please avoid to use the references in the abstract.

Page 1 – Lines from 26 to 30. I suggest to add in these sentences the Research Question and the specific aims to allow readers to see the basis of your narrative review.

Page 1 – Line 35. I suggest to insert here a brief introduction of your narrative review. This is important for the readers to be clear what the practical question is that you are trying to address. Please try to clarify why we need this article. The key issue here is to make you sure to set-up a correct approach to the review. I would suggest also to define better what are the primary outcome and the secondary outcomes of your review.

Page 3 – Lines from 136 to 139. In this sentence there are a lot of abbreviations that are not fully specified. I would suggest to write them fully since there are the first time that they appear in the text. I would also suggest to insert the “List of Abbreviations “ at the beginning of the manuscript.

Page 4 – Line 194. Please avoid to use abbreviation for the title.

Page 5 – Line 239. Please avoid to use abbreviation for the title.

Page 6 – Line 282. Please avoid to use abbreviation for the title.

Page 7 – Line 309. Please avoid to use abbreviation for the title.

Page 9 – Line 439. I would suggest to insert here a paragraph titled “Conclusions”, in which you could add what are the practical applications of your study. What should now physicians, trainers and practitioners now have to do after reading your paper? Based on the evidence-based information that you reported does it affect practice?

Author Response

Reviewer 2

I would like to thank the Editor for the opportunity to review this manuscript for International Journal of Environmental Research and Public Health. This is a narrative review that aimed to present the current scientific evidence on the effects of sub-concussive impacts in contact sports. I really appreciate the effort of the authors working in this field, because sport-related concussion and sub-concussions has a severe impact in teenager athletes’ health. This review is really well written and organized from a methodological point-of-view. Please, find my specific comments here below in order to improve the quality of the manuscript

Specific comments

  1. Page 1 – Line 20. Please avoid to use the references in the abstract.

Reply point 2: The reference in the abstract has been removed, also further to Reviewer 3’s comment (point 13). Line 19 of the abstract now reads: ‘recent studies discovered’.

  1. Page 1 – Lines from 26 to 30. I suggest to add in these sentences the Research Question and the specific aims to allow readers to see the basis of your narrative review.

Reply point 3: Thank you for this suggestion. Lines 24-29 (changes highlighted in red) of the abstract now read: The aim of this review is to highlight the current scientific evidence on the effects of repeated sub-concussive head impacts on contact sports athletes’ brains, identify the areas in need of further investigation, highlight the potential of advanced neuroscientific methods, and comment on the steps governing bodies have made to address this issue. We conclude that there are indeed neural and biofluid markers that can help understand better the effects of repeated sub-concussive head impacts and that some aspects of contact sports should be redefined, especially in situations where sub-concussive impacts and concussions can be minimized.

  1. Page 1 – Line 35. I suggest to insert here a brief introduction of your narrative review. This is important for the readers to be clear what the practical question is that you are trying to address. Please try to clarify why we need this article. The key issue here is to make you sure to set-up a correct approach to the review. I would suggest also to define better what are the primary outcome and the secondary outcomes of your review.

Reply point 4: Once again, thank you for this suggestion. We have now added a clearer outline of the narrative review under the heading ‘Introduction’ (lines 41-54). The new section is highlighted in red.

  1. Page 3 – Lines from 136 to 139. In this sentence there are a lot of abbreviations that are not fully specified. I would suggest to write them fully since there are the first time that they appear in the text. I would also suggest to insert the “List of Abbreviations “at the beginning of the manuscript.

Reply point 5: We agree with the reviewer and therefore the use of abbreviations has been eliminated as much as possible throughout the entire manuscript. All neuroimaging terms in the section have now been spelled out and recurring abbreviations in the manuscript are now included in a list of abbreviations as follows (please note that a qMRI section has been added, highlighted in red):

Abbreviations: RSHI = Repetitive Sub-concussive Head Impact; CTE = Chronic Traumatic Encephalopathy; MRI = Magnetic Resonance Imaging; qMRI = quantitative Magnetic Resonance Imaging; MRS = Magnetic Resonance Spectroscopy; TMS = Transcranial Magnetic Stimulation; EEG = electroencephalogram.

The journal has moved these to a section at the end of the manuscript (lines 504-507).

  1. use abbreviation for the title:

Page 4 – Line 194. Please avoid to use abbreviation for the title.

Page 5 – Line 239. Please avoid to use abbreviation for the title.

Page 6 – Line 282. Please avoid to use abbreviation for the title.

Page 7 – Line 309. Please avoid to use abbreviation for the title.

Reply point 6: All use of abbreviation in any of the titles has been removed.

  1. Page 9 – Line 439. I would suggest to insert here a paragraph titled “Conclusions”, in which you could add what are the practical applications of your study. What should now physicians, trainers and practitioners now have to do after reading your paper? Based on the evidence-based information that you reported does it affect practice?

Reply point 7: A new section ‘Conclusions’ (line 480-495) has now been included in the revised manuscript (highlighted in red).

Reviewer 3 Report

The authors have submitted a paper on the negative neurological aspects of contact sports. The paper is well structured and of adequate length for the topic. However, several improvements should be brought:

  • In section 1, instead of the general recommendations for sports activities, which are widely popularized even in the non-medical population, perhaps a better fit would be a summary of proven positive effects, citing studies demonstrating health-promoting effects (aging, cardiovascular status, cognition, and others).
  • Chapter 2 seems to focus only on concussions, so the title should be adjusted. Alternatively, if the title is to be kept, all the other negative effects should be presented
  • More information needs to be provided in Chapter 3. There are plenty of studies on American football players, boxers, and other sports that have registered the negative aspects of concussions as early as 30 years ago - extensive data is available on the cognitive features in these athletes.
  • Chapter 4 focuses on the biomarkers evaluated by various methods. A table or a figure would help improve the readability of this section and summarize all the various imaging and paraclinical markers.
  • Subheading 4.1. Neuroimaging seems to be improper as subsections 4.2., 4.3 and forward seem to be subdivisions of Neuroimaging. This should be reorganized.
  • Since no references may be introduced in the abstract, perhaps it is not advised to cite other studies (i.e. Magdalena Ietswaart's). An alternative would be to mention that "other/recent studies have shown..."
  • Some references need to be updated according to the instructions for the authors

Respectfully submitted,

Author Response

Reviewer 3

Comments:

The authors have submitted a paper on the negative neurological aspects of contact sports. The paper is well structured and of adequate length for the topic. However, several improvements should be brought:

  1. In section 1, instead of the general recommendations for sports activities, which are widely popularized even in the non-medical population, perhaps a better fit would be a summary of proven positive effects, citing studies demonstrating health-promoting effects (aging, cardiovascular status, cognition, and others).

Reply point 8: We have firmed up section 1 of the manuscript to make it more targeted with regards to the focus of the review, also further to the comments of Reviewer 2. With this, the popularized recommendations for sports activities that Reviewer 3 refers to no longer feature in the revised manuscript. 

  1. Chapter 2 seems to focus only on concussions, so the title should be adjusted. Alternatively, if the title is to be kept, all the other negative effects should be presented

Reply point 9: The title of this section has been adjusted, following the comments of Reviewer 3, and the relevance of concussion in the context of the aims of the review (now more clearly set out also further to the comments of Reviewer 2) has been clarified in section 1 and 2 of the revised manuscript, as highlighted in red.

  1. More information needs to be provided in Chapter 3. There are plenty of studies on American football players, boxers, and other sports that have registered the negative aspects of concussions as early as 30 years ago - extensive data is available on the cognitive features in these athletes.

Reply point 10: We agree with the reviewer that the title of this section was misleading. The title and the text in this section as well as the abstract and the introduction (changes are highlighted in red) has been adjusted to make clear that this section is not so much about concussion, also further to comments 3 & 4 of Reviewer 2, but instead about what cognitive markers tell us about the consequences of sub-concussive (routine) impact in sport.

  1. Chapter 4 focuses on the biomarkers evaluated by various methods. A table or a figure would help improve the readability of this section and summarize all the various imaging and paraclinical markers.

Reply point 11: Further to this reviewer’s point regards to the organisation of this section (point 12 below) and following removal of the use of acronyms that impeded the readability of the manuscript for those not familiar with these methods, we hope that the readability of this section has much improved. As this is a narrative review (also see Reviewer 2), we worry that a figure or table may not improve the readability of this section as we attempt to discuss the evidence in context and guide the reader through narrative in light of the aims of this review (which we hope are now more clearly set out in the revised manuscript throughout, see additional text highlighted in red).

  1. Subheading 4.1. Neuroimaging seems to be improper as subsections 4.2., 4.3 and forward seem to be subdivisions of Neuroimaging. This should be reorganized.

Reply point 12: Improper subsections have been addressed.

  1. Since no references may be introduced in the abstract, perhaps it is not advised to cite other studies (i.e. Magdalena Ietswaart's). An alternative would be to mention that "other/recent studies have shown..."

Reply point 13: This reference in the abstract has been removed, also further to Reviewer 2’s comment (point 2). Line 19 of the abstract now reads: ‘recent studies discovered’.

  1. Some references need to be updated according to the instructions for the authors

Reply point 14: References have been updated according to the instructions for authors.

Round 2

Reviewer 2 Report

Thank you very much for addressing my previous comments. I think the paper is much improved, and you should be commended for your thorough approach to the editing process.

One small comment, that you might consider in this paragraph is regarding the potentially negative effects of sports on brain health. In a recent systematic review and meta-analysis regarding the incidence of pediatric and adolescent concussion in action sports was reported that m
otocross, sailing and snowboarding presented the highest incidence rates per 1000 athlete exposure at 39.22, 3.73 and 2.77 respectively, whereas alpine skiing had the lowest incidence rates resulting in 0.30. Overall risk of concussion was estimated at 0.33 (CI: 0.22, 0.45). However, analysing the articles authors concluded that a significant limitation in ananlysing the studies was the lack of a definition of concussion in many of them. The diagnosis of concussion may be difficult as the symptoms may be subtle, and in many cases, a high level of suspicious is required for the diagnosis, thus resulting in underestimation of the number of concussion. It is also possible that some studies generically included head trauma or injuries, which fall outside the true definition of concussion. Furthermore, they did not address the influence of study quality on the results because many data were not reported in many studies, For example, the stratification in respect to age was not reported in most series, while critical data such as the previous experience of concussion, the follow up and the mechanism of injuries were generally not reported. Most studies failed to identify whether concussions occurred during competition or simple practice.

Author Response

Thank you for your comments. We have in fact undertaken an analysis of a large cross-Europe cohort of sport-related concussions (Ntikas et al., in preparation). As our findings will be quite different from the above, but as we are keen to take forward the points of this reviewer, we have amended the manuscript by including a short statement to this section (see section 2 of the revised manuscript, new text highlighted in red) to emphasising the importance of large-scale international cohort studies to examine the scale of the problem. This reads as follows:

However, the examination of the incidence of traumatic brain injury at scale dated back at least 20 years, until the launch of a recent large-scale international observational study [Maas et al. 2015]. Cohort studies like this are critical to representatively assess the scale of the problem: examining sport-related brain injury incidence, sport-specific profiles, demographics such as age and sex, and critically the incidence and outcomes of those incurring multiple brain injuries while participating in sport.

Reference added: doi.org/10.1227/neu.0000000000000575

Reviewer 3 Report

The authors have addressed all previously-reported concerns

Author Response

Thank you.